# How Well Do LLMs Identify Cultural Unity in Diversity?

**Jialin Li & Junli Wang**
Tongji University
`{2233032,junliwang}@tongji.edu.cn`

**Junjie Hu**
University of Wisconsin-Madison
`junjie.hu@wisc.edu`

**Ming Jiang**[*]
Indiana University Indianapolis
`mj200@iu.edu`

## Abstract

Much work on the cultural awareness of large language models (LLMs) focuses on the models' sensitivity to geo-cultural diversity. However, in addition to cross-cultural differences, there also exists common ground across cultures. For instance, a *bridal veil* in the *United States* plays a similar cultural-relevant role as a *honggaitou* in *China*. In this study, we introduce a benchmark dataset **CUNIT**[1] for evaluating decoder-only LLMs in understanding the cultural unity of concepts. Specifically, CUNIT consists of 1,425 evaluation examples building upon 285 traditional cultural-specific concepts across 10 countries. Based on a systematic manual annotation of cultural-relevant features per concept, we calculate the cultural association between any pair of cross-cultural concepts. Built upon this dataset, we design a contrastive matching task to evaluate the LLMs' capability to identify highly associated cross-cultural concept pairs. We evaluate 3 strong LLMs, using 3 popular prompting strategies, under the settings of either giving all extracted concept features or no features at all on CUNIT. Interestingly, we find that cultural associations across countries regarding clothing concepts largely differ from food. Our analysis shows that LLMs are still limited to capturing cross-cultural associations between concepts compared to humans. Moreover, geo-cultural proximity shows a weak influence on model performance in capturing cross-cultural associations.

## 1 Introduction

Recent advances in large language models (LLMs) have significantly empowered the machines' knowledge capacity in a variety of general domains such as math (Ouyang et al., 2022; Stolfo et al., 2023), logical reasoning (Liu et al., 2023b; Xu et al., 2023), and commonsense (Bang et al., 2023; Bian et al., 2023). While humans share common knowledge, they also possess diverse, community-specific knowledge, especially in cultural contexts. Inspired by that, recently, scholars have been actively exploring LLMs' cultural awareness (Li & Zhang, 2023; Huang & Yang, 2023; Wu et al., 2023; Hershcovich et al., 2022), to facilitate cultural knowledge dissemination and culturally-aware communication by LLMs.

Prior studies generally consider LLMs' cultural awareness from two perspectives: (1) linguistic-level variations caused by cultural diversity such as word usage (Shaikh et al., 2023) and language style (Kabra et al., 2023; Liu et al., 2023a), and (2) knowledge-level variations like social norms (Fung et al., 2023; Ziems et al., 2023a; CH-Wang et al., 2023) and cultural inferences(Liu et al., 2021; Yin et al., 2022; Li & Zhang, 2023) across geopolitical regions. Despite valuable contributions made by existing research, these studies primarily focus on examining the sensitivity of models to the connections between geopolitical regions and their associated cross-cultural concepts. The emphasis lies on the machine's awareness

---

[*]Corresponding author
[1]Our dataset and code are released publicly at `https://github.com/ljl0222/CUNIT`

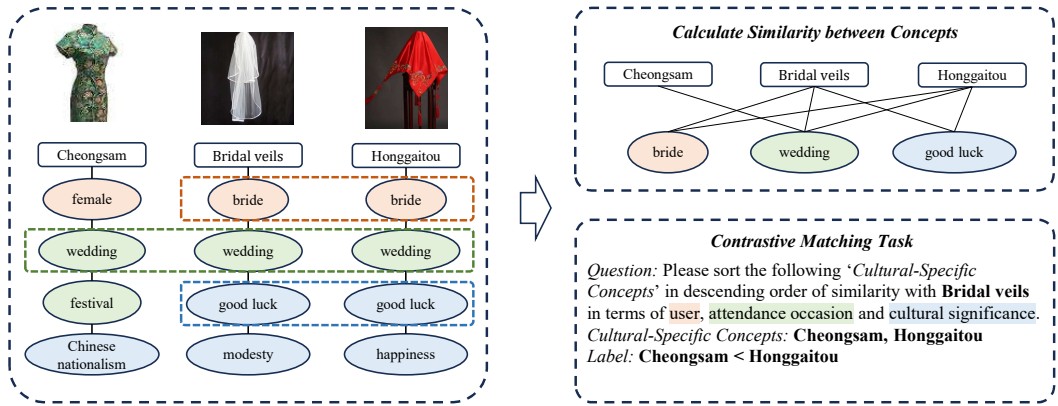

Figure 1: **Illustrative example of a CUNIT data instance for contrastive matching.** Given a culturally specific concept, "Bridal veils", from a query culture (e.g., United States) and two culturally specific concept candidates, "Cheongsam" and "Honggaitou", from the target culture (e.g., China), our goal is to ask an LLM to determine which target concept shares a higher cultural-centered similarity to the query concept. This comparison is based on the concepts' pragmatic features in three categories: users (e.g., bride), cultural-specific occasions (e.g., wedding), and cultural significance (e.g., good luck).

of geo-cultural diversity. However, there is scant research exploring the capacity of LLMs to grasp culturally centered associations among concepts specific to diverse geo-cultures. Differing from prior studies that highlight LLMs' awareness of geo-cultural diversity, this alternative perspective shifts its focus towards investigating the models' potential to capture shared aspects across cultures. The goal of this paper is to assess LLMs' competence in facilitating intercultural communication.

In this study, we evaluate LLMs' ability to align cross-cultural concepts by their inherent cultural-centered associations. Although language and culture are highly intertwined, we focus on LLMs pre-trained on English-dominant corpora. This is mainly because English is the most accessible language, with extensive data encompassing massive global knowledge, including concepts from diverse cultures. Specifically, we concentrate on cultural-centered association in three aspects: (1) the social group of the cultural object's users, (2) cultural-specific occasions, and (3) cultural significance. Figure 1 shows an illustrative example. The object "Honggaitou" is a traditional bridal veil worn by the Han Chinese brides at their wedding ceremony, like "Bridal veils" in Western culture. Although both items differ in clothing shape and materials, they are used in a similar cultural scenario (i.e., worn by brides at their weddings) and share the same cultural significance (e.g., symbolizing good fortune). Therefore, the concept "Bridal veils" should have more cultural equivalence with the concept "Honggaitou" compared with "Cheongsam" (i.e., a traditional Chinese dress worn by women in special events like wedding ceremonies). This new paradigm of accessing LLMs' cultural awareness will empower machines' mutual understanding of diverse cultural knowledge, benefiting cross-cultural alignment in downstream applications like machine translation (Yao et al., 2023) and multimodal reasoning (Li & Zhang, 2023).

To conduct this study, we design a new contrastive matching task and introduce a novel benchmark **CUNIT**. Specifically, we curate a high-quality set of cross-cultural concepts with detailed feature annotations over 10 countries, and create question-answering pairs from the curated concepts to evaluate LLMs on identifying concept pairs with a higher cultural similarity. We perform an in-depth probing analysis of three popularly used decoder-based LLMs by utilizing popular prompting strategies. Considering the potential challenges that the frequency of cross-cultural concepts might pose for LLMs in deducing implicit cultural-centered associations, we evaluate LLMs in two settings with or without providing external features of cultural concepts in the prompt to LLMs.

Overall, the GPT model outperforms the open-sourced LLaMA model in terms of prediction accuracy and consistency in both settings. We further investigate the impact of the cross-cultural concepts' frequency and geographical locations on model predictions. Our results reveal that LLMs are still limited to capturing the cultural similarity between low-frequency concepts in the long tail. Besides, LLMs can benefit from additional culture-related annotations in the prompt when comparing two concepts from geographically distant cultures.

## 2 Related Work

**Knowledge probing.** Advances in pre-trained language models (PLMs) have spurred a variety of probing designs to uncover embedded facets of the world in these models (Youssef et al., 2023; Belinkov, 2022). Popular topics include linguistic properties (Shaikh et al., 2023; Kabra et al., 2023; Liu et al., 2023a), biases (Hendrycks et al., 2021; Parrish et al., 2022; Huang & Xiong, 2023), facts (Petroni et al., 2019; Kassner et al., 2021; Jiang et al., 2020), and commonsense knowledge (Bang et al., 2023; Ouyang et al., 2022)

Early work typically views probing as a classification problem, where a classifier is built on top of the intermediate representations of selected PLMs to predict target properties (Baroni et al., 2014; Conneau et al., 2018; Tenney et al., 2019). With the rise of encoder-based PLMs pre-trained by masked language modeling, the cloze-style probing has been widely used to examine the models' knowledge capacity (Zhong et al., 2021; Yu et al., 2023). Recently, following the generative nature of decoder-based PLMs, especially LLMs, the design of probing in a question-answering (QA) format has become popular (Singhal et al., 2022; Tack & Piech, 2022; Blair-Stanek et al., 2023). Our study falls into this probing group and focuses on LLMs regarding a new probing perspective—geo-cultural competence.

**Cultural awareness of language models.** Research in language models' awareness of cultural factors has received increasing attention in the NLP community (Jiang & Joshi, 2024; Ramezani & Xu, 2023; Jha et al., 2023). Existing work in this field covers a variety of explorations, ranging from cross-cultural differences in word usage (Shaikh et al., 2023; Li et al., 2023; Hu et al., 2023) to dialect-associated biases in PLMs (Kabra et al., 2023; Ziems et al., 2023b; Sun et al., 2023), and from the correlation between cross-cultural shifts and cross-lingual shifts (Yao et al., 2023), to geo-diverse commonsense reasoning evaluation (Yin et al., 2022). Despite various investigations, the majority of prior studies emphasize the sensitivity of language models to cultural diversity regarding geo-political regions.

Differing from prior work concentrating on the models' awareness of geo-cultural differences, we are interested in the models' potential to capture latent common ground across cultures. To the best of our knowledge, only one latest work (Li & Zhang, 2023) has touched on this topic, where the authors propose an automatic method to align any low-resource cross-cultural concept with a high-resource cross-cultural concept based on their shared semantic category attributes annotated on WordNet. Compared with Li & Zhang (2023) that considers this alignment problem from an on-surface semantic perspective, we emphasize culture-centered pragmatic-level associations among cross-cultural concepts. With this emphasis, our ultimate goal is to measure the geo-cultural competence of LLMs.

## 3 CUNIT Benchmark for Cultural Contrastive Matching

### 3.1 Task Definition

We introduce a contrastive matching task to evaluate LLMs' capability to capture concept-level cultural similarity in different geo-cultures. Specifically, given a triplet of cultural-specific concepts $(c^q, c_1^t, c_2^t)$, where $c^q$ comes from a query culture $q$, $c_1^t$ and $c_2^t$ come from a target culture $t$, we evaluate an LLM on identifying a concept from $c_1^t$ and $c_2^t$ that shares a higher cultural similarity with the query $c^q$ in terms of three pragmatic categories $(g_1, g_2, g_3)$. We formulate this task as a generative question-answering format, where an LLM is asked to generate a text answer to the question like "Please sort the following cultural-specific concepts in descending order of similarity with $c_q$ in terms of $g_1, g_2, g_3$. Cultural-specific

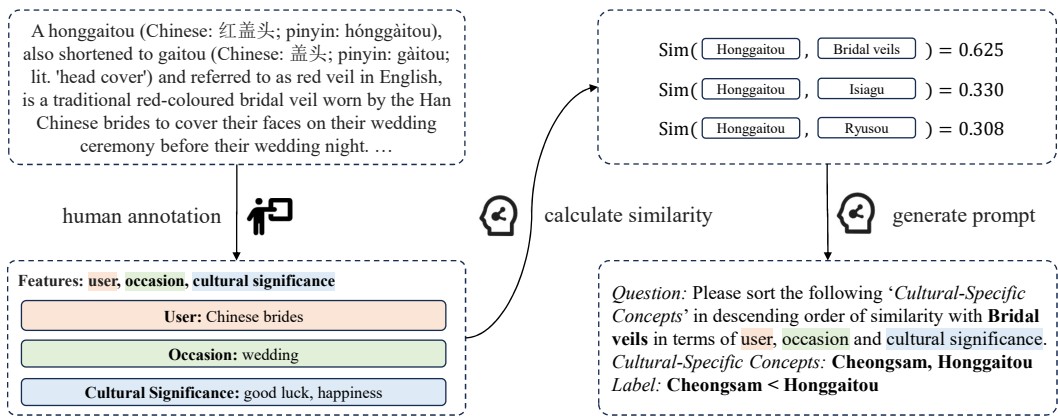

Figure 2: **The pipeline of CUNIT construction.** (1) Find relevant descriptions of cultural-specific concepts from Wikipedia. (2) Extract cultural-relevant features of each concept, and map them into a unified feature schema towards manual annotation (e.g., using *'bride'* to represent synonym features like *'Chinese brides'*, *'brides'*). (3) Calculate the cultural similarity between any pair of cross-cultural concepts. (4) Construct testing cases by different prompt strategies for LLM evaluation.

concepts: $c_1^t$, $c_2^t$." Detailed prompting examples are shown in Table 6,7,8 in Appendix D. Following the standard practice of prior work (Hossain et al., 2023; Li & Zhang, 2023), we categorize different cultures based on their geo-locations at the country level.

## 3.2 CUNIT Data Curation

To examine LLMs' performance on this contrastive matching task, we construct a benchmark dataset called CUNIT. Figure 2 displays an overview of our data curation pipeline.

**Cross-Cultural Concept Collection.** Our pipeline starts by collecting cross-cultural concepts and their descriptions from Wikipedia.[2] In this study, we narrow our focus to two material cultural categories: clothing and food(more details shown in Table 4 in Appendix A). This selection is based on our preliminary statistics of accessible cross-cultural concepts across various cultural-relevant categories (e.g., architecture, performing arts) that demonstrate strong cross-cultural associations.

To ensure geo-culture diversity, we select 10 countries across 5 continents[3] that have a large number of cross-cultural concepts in the two selected categories(shown in Table 1). To further investigate the influence of geo-cultural proximity on the models' capability to capture cultural similarity, with an emphasis on comparing Eastern-centric cultures versus Western-centric cultures, we considered more Asian countries compared to others.

**Feature Annotation for Cross-Cultural Concepts.** Motivated by prior studies in intercultural communication (Wiseman, 2003; Wang et al., 2006), we aim to measure the cultural similarity between any pair of cross-cultural concepts originating from two different cultures. Specifically, we focus on comparing the pragmatic information between two cross-cultural concepts, especially examining the context of usage in their cultures. As such cultural nuance is hard to extract from pre-trained word embeddings of each concept, we aim to construct a list of categorical pragmatic features for each cross-cultural concept, covering three pre-defined feature categories: (1) the social group of the cultural concept's users,

---

[2]Clothing: `https://en.wikipedia.org/wiki/Category:Clothing_by_country` and Food: `https://en.wikipedia.org/wiki/Category:Cuisine_by_country`.

[3]We exclude South America due to the limited availability of its cross-cultural concepts on Wikipedia.

| Category | China | India | Japan | Korea | Thailand | Australia | America | Mexico | Italy | Nigeria | All |
|---|---|---|---|---|---|---|---|---|---|---|---|
| Clothing | 25 | 12 | 20 | 24 | 5 | 7 | 19 | 5 | 24 | 6 | 153 |
| Food | 17 | 17 | 9 | 3 | 2 | 14 | 30 | 4 | 31 | 5 | 132 |
| All | 42 | 29 | 29 | 27 | 7 | 21 | 49 | 11 | 55 | 9 | 285 |

Table 1: **Countries and cultural concepts in the benchmark.** Our benchmark CUNIT for evaluation includes a total of 10 countries. We only consider cross-cultural concepts with more detailed descriptions in Wikipedia in the process of constructing CUNIT.

such as gender, social class, and job; (2) the cultural-specific occasion of the concept, such as wedding ceremonies, festivals, and workplace; and (3) the cultural significance of the concept, such as national identity, happiness, and wealth.

Notably, we construct this feature list while annotating our collected concepts. Specifically, to avoid intensive manual annotation yet guarantee the annotation quality, for each concept, we start by designing a simple prompt(shown in Appendix D) to instruct ChatGPT[4] to extract a list of excerpts from the concept's Wikipedia article that are relevant to each feature category. Based on the extracted excerpts for each concept, we manually extract the key phrases that contain pragmatic information in our three pre-defined feature categories, and use those key phrases as candidate features. To validate the model's performance in extracting excerpts, we conducted a preliminary analysis by having an annotator review ChatGPT's outputs for 50 randomly selected concepts using their full Wikipedia articles, achieving a recall of 94.42%. To avoid mis-annotations of concepts due to limited texts, we further filter out concepts that contain key phrases in only one feature category. After processing all concepts' Wikipedia articles, we obtain a set of key phrases that potentially contain semantically similar phrases due to paraphrasing. Therefore, to mitigate feature sparsity, we manually conducted a phrase normalization that converts multiple synonymous key phrases into a single unique term. We ask two human annotators to extract key phrases and conduct the same phrase normalization to create two feature lists, and obtain a Kappa coefficient of 0.9391 for the two annotators, suggesting a strong agreement. Finally, we manually merge the two feature lists into a list of 164 categorical features (including 98 features for clothing and 66 features for food), and each concept has 4~5 annotated cultural-relevant features on average. Appendix B shows the full list of features.

As each concept is annotated with categorical features, we use the Jaccard similarity to calculate the cultural similarity between any pair of cross-cultural concepts $(c_i, c_j)$:

$$\text{Sim}(c_i, c_j) = \frac{|F_{c_i} \cap F_{c_j}|}{|F_{c_i} \cup F_{c_j}|},$$

(1)

where $F_{c_i} = \{f_{c_i}^1, f_{c_i}^2, f_{c_i}^3, ...\}$ and $F_{c_j} = \{f_{c_j}^1, f_{c_j}^2, f_{c_j}^3, ...\}$ denote the sets of the annotated features associated with the concepts $c_i$ and $c_j$ respectively.

**Sampling Testing Triplets for Evaluation.** Given the annotated cross-cultural concepts, we aim to create a set of meaningful concept triplets $\mathcal{D} = \{(c^q, c_1^t, c_2^t)_u\}_{u=1}^U$ for the contrastive matching task defined in Section 3.1. As cultural similarity varies largely across cross-cultural concept pairs, randomly sampling a concept triplet $(c^q, c_1^t, c_2^t)$ is likely to result in incomparable cases where both $c_1^t$ and

| Granularity | Sim. Difference | Clothing | Food |
|---|---|---|---|
| Large | $(\mu + 0.5\sigma, 1]$ | 231 | 156 |
| Middle | $(\mu - 0.5\sigma, \mu + 0.5\sigma]$ | 221 | 230 |
| Small | $[0, \mu - 0.5\sigma]$ | 248 | 339 |

Table 2: Concept triplets in three different granularities of similarity difference.

$c_2^t$ both have a very low cultural similarity with the query concept $c^q$. To address this issue, we sample concept triplets satisfying the constraint that one candidate concept has a cultural similarity with the query concept $c^q$ larger than 0.5, whereas the other candidate concept has a similarity smaller than 0.5. Overall, we have a set of 1,425 testing triplets in total, including 700 clothing triplets and 725 food triplets.

---

[4] We used the model gpt-3.5-turbo-0613 in this study.

To provide a fine-grained analysis of LLM performance on $\mathcal{D}$, we further count the distribution of the similarity difference between the pair of $(c^q, c_1^t)$ versus $(c^q, c_2^t)$ across all triplets, i.e., $\text{Sim}(c^q, c_1^t) - \text{Sim}(c^q, c_2^t)$, $\forall (c^q, c_1^t, c_2^t) \sim \mathcal{D}$, and compute the mean $\mu$ and standard deviation $\sigma$ of the similarity difference. We then group the triplets with their similarity differences falling within or beyond $0.5\sigma$ from $\mu$, indicating a large, middle, or small level of similarity difference. A small level of similarity difference indicates a harder contrastive matching task, which allows us to evaluate LLMs to tackle difficult cultural triplets. Table 2 shows the statistics of cross-cultural concept triplets per category at each level of granularity.

## 3.3 CUNIT Data Analysis

**Cross-cultural Unity.** Considering that regional and continental variations result in differences in the distribution of features associated with cross-cultural concepts, we calculate the average similarity between all pairs of collected concepts from any two different countries (see Figure 3). Interestingly, we observe a clear similarity cluster among concept pairs from two geolocationally close countries, e.g., clothing concepts among Asian countries.

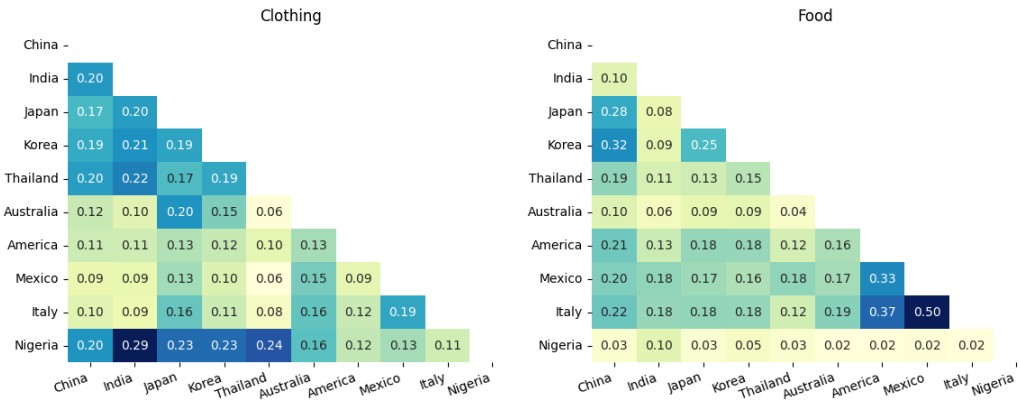

Figure 3: **Average similarity of concept pairs between different countries.** We calculate the similarity of different concepts between countries.

**Long-tail Distribution of Cross-Cultural Concepts Frequency.** We hypothesize that our collected cross-cultural concepts also follow a long-tail distribution which may affect the LLM performance on the cultural contrastive matching task. To verify this hypothesis, we use the Google search engine to estimate the frequency of each curated concept. Specifically, we first concatenate the text string of a concept with its corresponding country to form a query as $Query = (concept, country)$, and then count the number of returned webpages as the approximated frequency for the concept. Indeed, we observe a long-tail distribution of the curated concepts in Figure 7 in Appendix C.

## 4 Experimental Setting

**Prompting Strategies.** Considering the sensitivity of LLM performance to prompts, we employ three popularly used prompting strategies to investigate LLMs' awareness of cultural unity between cross-cultural concepts, including (1) the input-output prompt (**IO**) in the template of "*Question*: [...] \n *Answer*: [...]"; (2) the one-shot prompt (**One-shot**) that provides an input-output exemplar following the target question; and, (3) the chain-of-thought (**CoT**) prompt that additionally gives a rationale of the answer in the selected exemplar, aiming to guide the model to generate a rational for the target question.

To investigate how much intrinsic knowledge of cultural unity is learned by LLMs during their pre-training stage, we apply the aforementioned prompting strategies in two evaluation settings with or without providing cultural-relevant features of the concepts in the prompt.

| Category | Model | Large | | | Middle | | | Small | | |
|---|---|---|---|---|---|---|---|---|---|---|
| | | IO | One-shot | CoT | IO | One-shot | CoT | IO | One-shot | CoT |
| | | | | | **With features** | | | | | |
| *Clothing* | human | $100.0_{\pm0.00}$ | — | — | $95.93_{\pm0.00}$ | — | — | $79.44_{\pm0.00}$ | — | — |
| | gpt-3.5 | $50.14_{\pm0.40}$ | $\mathbf{52.31}_{\pm0.53}$ | $\underline{58.37}_{\pm0.89}$ | $54.82_{\pm0.00}$ | $55.88_{\pm0.56}$ | $\underline{\mathbf{60.78}}_{\pm1.30}$ | $46.03_{\pm0.57}$ | $44.69_{\pm0.50}$ | $\underline{52.41}_{\pm0.50}$ |
| | llama-13b | $41.99_{\pm0.00}$ | $49.13_{\pm0.00}$ | $\underline{49.78}_{\pm0.00}$ | $38.01_{\pm0.00}$ | $\underline{49.32}_{\pm0.00}$ | $48.87_{\pm0.00}$ | $40.32_{\pm0.00}$ | $\underline{\mathbf{50.00}}_{\pm0.00}$ | $49.60_{\pm0.00}$ |
| | llama-7b | $36.80_{\pm0.00}$ | $38.10_{\pm0.00}$ | $\underline{47.62}_{\pm0.00}$ | $30.32_{\pm0.00}$ | $39.37_{\pm0.00}$ | $\underline{\mathbf{50.00}}_{\pm0.00}$ | $37.50_{\pm0.00}$ | $41.33_{\pm0.00}$ | $\underline{48.39}_{\pm0.00}$ |
| *Food* | human | $94.87_{\pm0.00}$ | — | — | $81.74_{\pm0.00}$ | — | — | $76.70_{\pm0.00}$ | — | — |
| | gpt-3.5 | $\mathbf{65.81}_{\pm0.30}$ | $65.92_{\pm0.60}$ | $\underline{67.20}_{\pm0.30}$ | $52.32_{\pm0.41}$ | $\underline{\mathbf{57.10}}_{\pm0.71}$ | $51.81_{\pm1.34}$ | $\underline{43.66}_{\pm0.00}$ | $37.81_{\pm0.37}$ | $36.68_{\pm0.77}$ |
| | llama-13b | $46.47_{\pm0.00}$ | $\underline{49.68}_{\pm0.00}$ | $45.51_{\pm0.00}$ | $\underline{48.26}_{\pm0.00}$ | $47.17_{\pm0.00}$ | $46.52_{\pm0.00}$ | $\underline{\mathbf{46.90}}_{\pm0.00}$ | $46.46_{\pm0.00}$ | $44.40_{\pm0.00}$ |
| | llama-7b | $40.71_{\pm0.00}$ | $50.32_{\pm0.00}$ | $\underline{52.24}_{\pm0.00}$ | $38.48_{\pm0.00}$ | $38.26_{\pm0.00}$ | $\underline{\mathbf{50.00}}_{\pm0.00}$ | $42.63_{\pm0.00}$ | $41.74_{\pm0.00}$ | $\underline{\mathbf{47.05}}_{\pm0.00}$ |
| | | | | | **Without features** | | | | | |
| *Clothing* | human | $93.51_{\pm0.00}$ | — | — | $88.69_{\pm0.00}$ | — | — | $66.53_{\pm0.00}$ | — | — |
| | gpt-3.5 | $48.27_{\pm0.74}$ | $45.74_{\pm0.89}$ | $\underline{50.72}_{\pm0.41}$ | $50.83_{\pm0.56}$ | $\underline{\mathbf{54.45}}_{\pm0.56}$ | $52.94_{\pm1.40}$ | $48.39_{\pm0.33}$ | $\underline{50.07}_{\pm1.56}$ | $46.77_{\pm1.06}$ |
| | llama-13b | $46.10_{\pm0.00}$ | $\underline{\mathbf{51.73}}_{\pm0.00}$ | $48.26_{\pm0.00}$ | $46.61_{\pm0.00}$ | $\underline{51.13}_{\pm0.00}$ | $47.51_{\pm0.00}$ | $46.37_{\pm0.00}$ | $49.80_{\pm0.00}$ | $\underline{\mathbf{50.20}}_{\pm0.00}$ |
| | llama-7b | $\underline{40.04}_{\pm0.00}$ | $26.62_{\pm0.00}$ | $37.45_{\pm0.00}$ | $\underline{39.14}_{\pm0.00}$ | $29.86_{\pm0.00}$ | $36.65_{\pm0.00}$ | $\underline{40.32}_{\pm0.00}$ | $31.65_{\pm0.00}$ | $39.72_{\pm0.00}$ |
| *Food* | human | $91.67_{\pm0.00}$ | — | — | $64.35_{\pm0.00}$ | — | — | $67.55_{\pm0.00}$ | — | — |
| | gpt-3.5 | $\underline{\mathbf{58.55}}_{\pm0.30}$ | $51.39_{\pm0.60}$ | $49.36_{\pm0.91}$ | $\underline{\mathbf{68.77}}_{\pm0.00}$ | $53.04_{\pm0.36}$ | $50.72_{\pm1.63}$ | $\underline{54.23}_{\pm0.37}$ | $49.90_{\pm0.37}$ | $51.87_{\pm0.87}$ |
| | llama-13b | $\underline{48.08}_{\pm0.00}$ | $37.50_{\pm0.00}$ | $46.47_{\pm0.00}$ | $50.22_{\pm0.00}$ | $\underline{\mathbf{57.61}}_{\pm0.00}$ | $57.17_{\pm0.00}$ | $48.53_{\pm0.00}$ | $48.67_{\pm0.00}$ | $48.08_{\pm0.00}$ |
| | llama-7b | $42.31_{\pm0.00}$ | $41.03_{\pm0.00}$ | $\underline{46.15}_{\pm0.00}$ | $40.65_{\pm0.00}$ | $31.96_{\pm0.00}$ | $\underline{46.30}_{\pm0.00}$ | $43.66_{\pm0.00}$ | $36.73_{\pm0.00}$ | $\underline{46.31}_{\pm0.00}$ |

Table 3: **Accuracy of three prompting strategies with or without providing features in the cultural contrastive matching task.** The **bold number** indicates the best-performing model in the same strategy, and underline number indicating the best-performing strategy in the same model.

Specifically, in the first setting (**W/ Features**), we list cultural-relevant features per concept in the prompt, whereas in the second setting (**W/O Features**), we directly ask the question in the prompt without providing any concept features. To further disentangle the influence of the curated concepts' long-tail distribution property on LLMs, we replace the concept mention with an artificial phrase (e.g., *concept A*, *concept B*), to see if a model can identify similar concepts only based on their features.

**Models.** We use 3 popular LLMs for evaluation: (1) **GPT-3.5** (GPT-3.5-turbo-0613, 175B parameters) which shows a higher quality in handling complex instructions than prior GPT-based models (Brown et al., 2020); (2) **LLaMA-7B** and (3) **LLaMA-13B** which are open-sourced LLMs pre-trained on the English-dominated corpora (Touvron et al., 2023). In addition to LLMs, we also employ 2 human annotators to answer questions with and without features respectively on CUNIT for comparison. The annotators' background is shown in Section 7.

**Metrics.** We use two evaluation metrics in this study. First, we measure the **accuracy** by checking if the LLM can correctly identify a more similar concept per concept triplet in the contrastive matching task. Second, we hypothesize that LLMs may have intrinsic biases in selecting the first or second candidate concepts regardless of the question. To assess such model biases, we evaluate the **consistency** of the LLM's prediction for the same question by flipping the order of the two candidate concepts in the prompt.

## 5 LLM Evaluation and Results

### 5.1 How well do LLMs identify similar cross-cultural concepts?

**Overall Accuracy.** Table 3 shows the accuracy results of humans and LLMs in identifying a cultural concept with a higher similarity in the cultural contrastive task. Overall, the accuracy of human significantly outperforms LLMs, indicating that LLMs still face challenges in identifying the latent cultural unity of concepts compared to humans. Comparatively, GPT-3.5 shows a higher prediction accuracy than LLaMA in most testing cases, suggesting that this closed-sourced model captures more common ground across cultures than LLaMA-based open-sourced models.

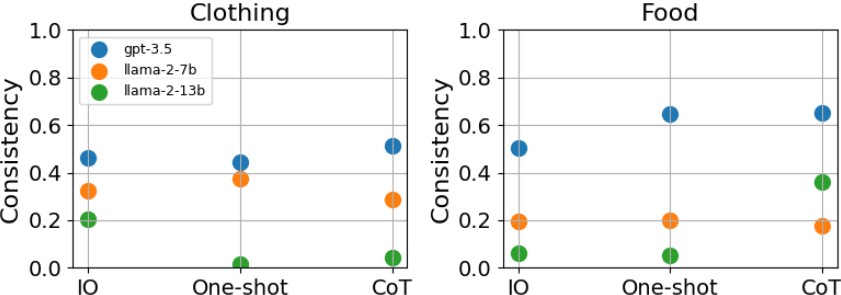

Figure 4: **Consistent performance across different models in experiments with features.**
Including the consistency performance of the three prompt strategies under gpt-3.5-turbo-0613, llama-2-7b-chat and llama-2-13b-chat, it can be found that gpt-3.5-turbo-0613 has the best consistency performance, significantly better than llama-2-7b-chat and llama-2-13b-chat.

**W/O Features.** We further compare the models' and humans' prediction accuracy in the evaluation setting without providing any concept features. Interestingly, unlike humans and GPT-3.5, which show improved prediction accuracy with prompts containing concept features, LLaMA variants, particularly LLaMA-13B, tend to perform more effectively when such features are absent from the prompts. One possible reason for this observation is that LLaMA is not trained robustly to utilize various forms of unseen information in the prompt at test time, and adding such unseen cultural features confuses the model.

**Contrastive Matching Granularity.** By further looking into the model accuracy among three levels of similarity difference between the candidates and the query concept, we find that humans tend to make more prediction errors when the similarity difference between two candidates with the query concept becomes smaller. However, this data factor does not play a clear effect on the models' predictions. We conjecture that although LLMs can identify a concept with a higher cultural similarity from a cultural concept triplet, these models cannot measure the numerical values of similarity accurately.

**Prompting strategies.** Regarding three prompting strategies, we find that CoT prompting generally guides models to achieve the highest prediction accuracy, followed by one-shot prompting and vanilla input-output prompting. Particularly, this pattern occurs more in the evaluation setting with features in the prompt. Our observation indicates that adding a chain-of-thought rationale and providing an exemplar both improve the models for identifying associated cross-cultural concept pairs.

**Stability of LLM prediction.** Given that prior studies have shown that LLMs may involve intrinsic biases to the order of candidate options in multi-choice question answering(Robinson & Wingate, 2023), we further investigate the consistency of LLMs' predictions by flipping the order of two candidate options. Figure 4 shows the consistency of the LLMs under each prompting strategy. We observe that GPT-3.5 obtains a noticeably higher consistency compared with LLaMA variants. This result indicates that the predictions of GPT-3.5 rely more on the semantic understanding of the questions rather than the order of the candidates.

## 5.2 How Do Data-centric Factors Affect LLMs' Predictions?

**Cultural Knowledge Representativeness.** Given that some cross-cultural concepts such as "Bridal veils" are much more well-represented than other concepts like "Honggaitou", we explore the sensitivity of LLM performance to cross-cultural concepts frequency. Specifically, we take the maximum frequency (based on Google Search Engine) of concepts in each triplet to denote the triplet's maximum representativeness. By sorting testing triplets by their representativeness, we compare the prediction accuracy of the best LLM (GPT-3.5) on the top 1/3 of well-represented triplets versus the bottom 1/3 of under-represented triplets. Figure 5

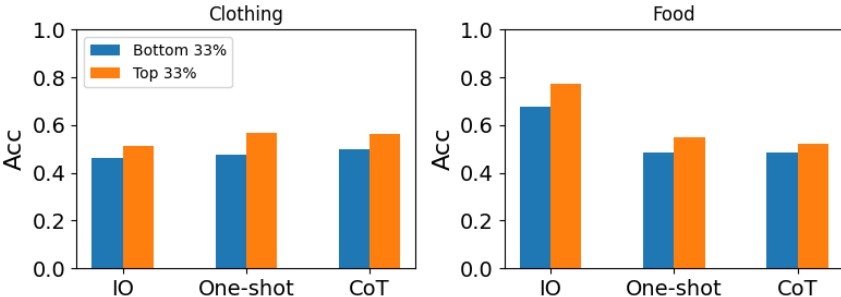

Figure 5: **The accuracy of GPT on concept triples(with features) in different long-tail degree groups.** We computed the maximum long-tail degree in each triplet, denoted as $d_{max}$. And we calculated the average accuracy of $d_{max}$ positioned within the first 1/3 and the last 1/3 of all triplets.

shows the prediction accuracy on clothing and food, respectively. We find that the model consistently exhibits a higher prediction accuracy on the well-represented triplets versus the under-represented ones across prompting strategies and cultural categories, indicating that long-tail cultural knowledge could challenge the model for identifying cross-cultural similarity.

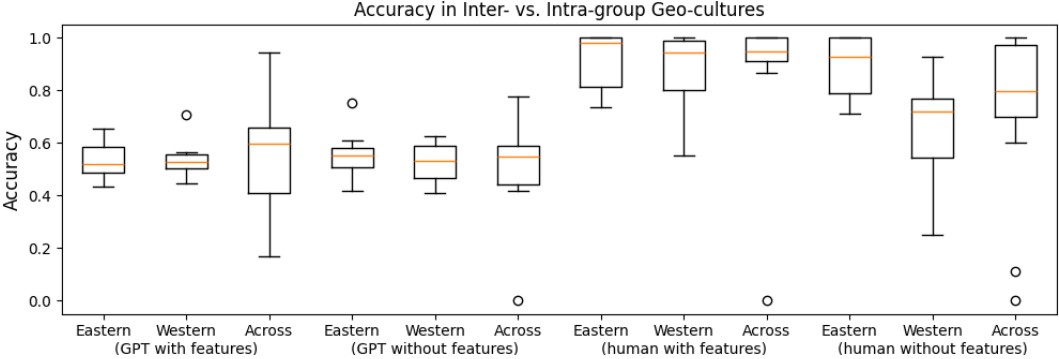

Figure 6: **The accuracy between different culture groups.** We compare accuracy of GPT and human when the concept triplets are from different culture groups. The length of the box represents the first quartile to the third quartile of the accuracy. The orange line shows the median of accuracy. The lines extending from the box extend to 1.5 times the IQR above and below the quartiles, and circles outside the box indicate obvious outliers

**Inter- vs. Intra-group Geo-cultural Proximity.** We further explore the model prediction accuracy to see if there exists geo-cultural proximity in the model's capability to identify similar cross-cultural concept pairs. We group the cultures into two groups: (1) Eastern and (2) Western cultures. Figure 6 is a boxplot showing the median (orange bar, with the first quartile to the third quartile in a box) of the accuracy in the cultural contrastive matching task, where we particularly focus on the model's performance against human annotators in the inter-/intra-group of Eastern-Western cultures. From the first three columns (GPT with features) in Figure 6, we observe a higher median accuracy (yet a larger variance) of GPT-3.5 on predictions of concept triplets across Eastern-Western cultures, compared to its predictions of concept triplets within the same cultures. However, in the setting without features, GPT-3.5 does not show a clear difference in terms of the median accuracy from inter- or intra-group of Eastern-Western cultures. Besides, human annotators outperform GPT-3.5 in terms of median accuracy in both settings with or without cultural features. When concepts originate from Eastern-centric cultures, annotators can provide more accurate predictions, as these annotators are from Eastern countries.

# 6 Conclusion

We propose a new benchmark CUNIT, and design a contrastive matching task to assess the cultural understanding of LLMs. The cross-cultural concepts collected by CUNIT demonstrate variations among different regions, with greater similarity observed between concepts among Eastern countries. Additionally, there is significant skewness in the long-tail distribution of different cross-cultural concepts. The contrastive matching task showcases the potential and consistency of GPT-3.5 in measuring cross-cultural unity, while open-sourced LLMs like LLaMA do not perform robustly to handle ordering and formatting issues. Meanwhile, the CoT prompting strategy demonstrates stronger prediction performance than vanilla input-output prompting and one-shot prompting. Regarding the data factors influencing cross-cultural similarity measures, we find that long-tail distribution of cross-cultural concepts and regional differences have an impact on models' prediction performance. Overall, LLMs still make more mistakes on long-tail concepts. In our preliminary work for cross-cultural research, we provide data on cross-cultural concepts and contrastive matching tasks, as well as analysis perspectives on long-tail degree and regional differences, which would facilitate future research in exploring the cross-cultural capabilities of LLMs.

# 7 Ethical Considerations

The construction of CUNIT involves two stages of human annotation. In the first stage, we focus on annotating concept features. Specifically, we employed 12 annotators from China, India, and the United States to identify culturally relevant features of each concept using Wikipedia descriptions (both the full text and ChatGPT-highlighted excerpts). The annotators are generally bilingual, with some being multilingual. The second stage involves contrastive matching on testing data. To establish upper-bound performance, we collected judgments from two additional annotators, both from China and proficient in English. Given the cultural ties in Asia, they possess sufficient cultural knowledge across Asian countries. We also provided a tutorial on our curated cultural concepts to further enhance their understanding of annotation. Despite that, there may still inevitably exist potential annotation biases given the annotator's cultural background.

# 8 Limitations

Considering that CUNIT is constructed on the basis of Wikipedia and specifically focuses on two types of material cultures across 10 countries, issues with data biases and representativeness are inevitable. Although we could collect data from sources beyond Wikipedia, maintaining data quality would be challenging. Another limitation is the limited cultural background of the annotators, which may affect their understanding of diverse cultural concepts, particularly unfamiliar ones, during annotation. To address these issues, a potential solution is to launch an open data curation platform that invites crowdsourcing volunteers to share their knowledge on cultural concepts, thereby expanding the valuable resources in CUNIT. In addition, our contrastive matching task is the first step in examining LLMs' understanding of cultural unity. In the future, we plan to extend our exploration to more practical downstream applications, such as machine translation.

# 9 Acknowledgements

We would like to thank the anonymous reviewers for their valuable feedback. MJ is partially supported by Google and the National Science Foundation (IIS-2438420).

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

## A  Preliminary Statistics

In Table 4, we give the number of cultural concepts of food and clothing in Wikipedia in different countries. The countries we select are the top-ranked countries in different continents. We also consider geographical proximity factors, so we select India, China, Japan, Korea, and Thailand within Asia.

## B  Features of Clothing and Food

In Table 5, we list the features in clothing and food categories used after normalization.

## C  Long-tail Distribution of Concepts

In Figure 7, we plot the distribution of webpage counts from the Google Search engine for all cross-cultural concepts, and we can see a clear long-tail pattern.

## D  CUNIT Data Examples

The prompt template used to filter concepts from Wikipedia descriptions is:

**Question:** Please extract sentences or phrases from the following context that can describe the cultural concept *Concept Name*, including the users of the cultural concept, the cultural occasions it is commonly used in, and its cultural significance.

**Context:** *Description*

In Table 6, 7, 8, we take Honggaitou, Cheongsam, and Bridal Veils as examples to show the prompt templates of different strategies we used. In the experiments without features, we removed the feature information of cultural concepts.

| Country | Continents | Clothing num | Food num | Sum |
|---|---|---|---|---|
| India | Asia | 351 | 209 | 560 |
| China | Asia | 143 | 195 | 338 |
| Japan | Asia | 94 | 239 | 333 |
| Mexico | America | 19 | 255 | 274 |
| Italy | Europe | 43 | 216 | 259 |
| France | Europe | 42 | 212 | 254 |
| Korea | Asia | 72 | 133 | 205 |
| Turkey | Asia | 32 | 172 | 204 |
| Germany | Europe | 38 | 146 | 184 |
| Britain | Europe | 67 | 111 | 178 |
| Indonesia | Asia | 44 | 134 | 178 |
| Pakistan | Asia | 80 | 98 | 178 |
| Iran | Asia | 59 | 104 | 163 |
| America | America | 87 | 75 | 162 |
| Greece | Europe | 35 | 115 | 150 |
| Spain | Europe | 21 | 120 | 141 |
| Thailand | Asia | 25 | 115 | 140 |
| Russia | Asia | 35 | 103 | 138 |
| Australia | Oceania | 46 | 89 | 135 |
| Poland | Europe | 12 | 116 | 128 |
| Vietnam | Asia | 23 | 100 | 123 |
| Bangladesh | Asia | 38 | 74 | 112 |
| Nepal | Asia | 13 | 99 | 112 |
| Portugal | Europe | 3 | 101 | 104 |
| Columbia | America | 5 | 94 | 99 |
| Ireland | Europe | 18 | 79 | 97 |
| Norway | Europe | 17 | 75 | 92 |
| Nigeria | Africa | 10 | 81 | 91 |
| Ukraine | Europe | 25 | 65 | 90 |
| Peru | America | 9 | 79 | 88 |
| Serbia | Europe | 5 | 83 | 88 |
| Netherlands | Europe | 15 | 72 | 87 |
| Algeria | Africa | 18 | 68 | 86 |
| Syria | Asia | 1 | 85 | 86 |
| Brazil | America | 1 | 84 | 85 |
| Finland | Europe | 4 | 79 | 83 |
| Canada | America | 5 | 77 | 82 |
| Austria | Europe | 6 | 75 | 81 |
| Chile | America | 8 | 70 | 78 |
| Palestine | Asia | 7 | 70 | 77 |
| Switzerland | Europe | 8 | 67 | 75 |

Table 4: **Number of clothing and food concepts in all countries collected from Wikipedia.** We calculate the quantity of concepts of country in Wikipedia categories 'Clothing by country' and 'Cuisine by country'.

| Type | Category | Features |
|------|----------|----------|
| Clothing | people | male, female, child, noble, commoner, emperor/empress, official, concubine, prince/princess, royal, politician, swagman, military, police, judge, farmer, labourer, frontiersman, athlete, postman, firefighter, rider, cowboy/cowgirl, fisher, trader, hunter, dancer, musician, servant, guard, scholar, monk, bride, married, unmarried |
| | occasion | formal, informal, ceremony, sacrifice, wedding, funeral, graduation, coming of age, worship ancestor, conferring, coronation, religious, gathering, court, political stage, casual, hunting, birthday, festival, celebration, military action, kayaking, drama, workplace, nightwear, swimming, labour, farming, fishing, horse-riding, riding |
| | meaning | hope, good luck, longevity, wealth, protect, fertility, happiness, morals, brave, virtue, spiritual, modesty, honor, social status, cultural heritage/symbol of nation, identity, unity, relationship, healing, marriage, religious beliefs, personality, emperor's favor, respect, pride, solidarity, revolution, welcome, nature, peace, rebellion, authority |
| Food | people | bride, groom, villager, tribe, military, child, stockman, noble, people related to religion |
| | occasion | ceremony, celebration, party, birthday, wedding, funeral, worship ancestor, feast, gathering, baptism, festival, Epiphany, Winter Solstice, Day of Dead, Passover, beginning of spring, Valentine's Day, Lantern Festival, Hannukah, Easter, Lebaran, Christmas, New Year, Hallows' Day, carnival holidays, Fool's Day, Children's Day, Mother's Day, Girl's Day, Shabbat, Thanksgiving, Deepavali, Purim, religious beliefs |
| | meaning | religious beliefs, happiness, wealth, hope, fertility, hospitalit, sharing, nature, bless, commemorate, pure, comfort, rural life, gratitude, respect, celebrate, reunion, health, longevity, beginning, good luck, waste, get rid of hardship |

Table 5: **Normalized feature words under different categories.** We list three feature words under different categories here: (1) the social group of the cultural object's users; (2) the cultural-specific occasion of the concept; and (3) the cultural significance of the concept.

| Strategy | Examples |
|----------|----------|
| Input-output Prompting w/o features | *Question:* Please sort the following 'Cultural-specific Concepts' in descending order of similarity feature overlap between 'Cultural-specific Concepts' with **Honggaitou** in terms of wearer, attendance occasion and symbolic meaning.
*Cultural-specific Concepts:* **Cheongsam**, **Bridal veils**
*Answer Format:* If **Honggaitou** and **Cheongsam** are more similar than **Honggaitou** and **Bridal veils** in terms of wearer, attendance occasion and symbolic meaning, please answer **Cheongsam** >**Bridal veils**, otherwise answer **Cheongsam** <**Bridal veils**.
*Answer:* |
| Input-output Prompting with features | *Question:* Please sort the following 'Cultural-specific Concepts' in descending order of similarity feature overlap between 'Cultural-specific Concepts' with **Honggaitou** in terms of wearer, attendance occasion and symbolic meaning.
*Cultural-specific Concepts:* **Cheongsam**, **Bridal veils**
*Features of Honggaitou:* 1. Wearer: female, bride; 2. Attendance occasion: wedding; 3. Symbolic Meaning: good luck, happiness
*Features of Cheongsam:* 1. Wearer: female; 2. Attendance occasion: wedding, festival; 3. Symbolic Meaning: China nationalism
*Features of Bridal veils:* 1. Wearer: female, bride; 2. Attendance occasion: wedding; 3. Symbolic Meaning: good luck, modesty
*Answer Format:* If **Honggaitou** and **Cheongsam** are more similar than **Honggaitou** and **Bridal veils** in terms of wearer, attendance occasion and symbolic meaning, please answer **Cheongsam** >**Bridal veils**, otherwise answer **Cheongsam** <**Bridal veils**.
*Answer:* |
| Input-output Prompting (w/o names) | *Question:* Please sort the following 'Cultural-specific Concepts' in descending order of similarity feature overlap between 'Cultural-specific Concepts' with **concept A** in terms of wearer, attendance occasion and symbolic meaning.
*Cultural-specific Concepts:* **concept B**, **concept C**
*Features of concept A:* 1. Wearer: female, bride; 2. Attendance occasion: wedding; 3. Symbolic Meaning: good luck, happiness
*Features of concept B:* 1. Wearer: female; 2. Attendance occasion: wedding, festival; 3. Symbolic Meaning: China nationalism
*Features of concept C:* 1. Wearer: female, bride; 2. Attendance occasion: wedding; 3. Symbolic Meaning: good luck, modesty
*Answer Format:* If **concept A** and **concept B** are more similar than **concept A** and **concept C** in terms of wearer, attendance occasion and symbolic meaning, please answer **concept B** >**concept C**, otherwise answer **concept B** <**concept C**.
*Answer:* |

Table 6: **Input-output prompt strategies' examples.** Each instance includes a prompt template for evaluating language models, the cultural-specific concepts and the required format for responses.

| Strategy | Examples |
|---|---|
| One-shot Prompting w/o features | *Question:* Please sort the following 'Cultural-specific Concepts' in descending order of similarity feature overlap between 'Cultural-specific Concepts' with Jeongjagwan in terms of wearer, attendance occasion and symbolic meaning.
*Cultural-specific Concepts:* Calceus, Pileus (hat)
*Answer Format:* If Jeongjagwan and Calceus are more similar than Jeongjagwan and Pileus (hat) in terms of wearer, attendance occasion and symbolic meaning, please answer Calceus >Pileus (hat), otherwise answer Calceus <Pileus (hat).
*Answer:* Calceus >Pileus (hat)

*Question:* Please sort the following 'Cultural-specific Concepts' in descending order of similarity feature overlap between 'Cultural-specific Concepts' with **Honggaitou** in terms of wearer, attendance occasion and symbolic meaning.
*Cultural-specific Concepts:* **Cheongsam**, **Bridal veils**
*Answer Format:* If **Honggaitou** and **Cheongsam** are more similar than **Honggaitou** and **Bridal veils** in terms of wearer, attendance occasion and symbolic meaning, please answer **Cheongsam** >**Bridal veils**, otherwise answer **Cheongsam** <**Bridal veils**.
*Answer:* |
| One-shot Prompting with features | *Question:* Please sort the following 'Cultural-specific Concepts' in descending order of similarity feature overlap between 'Cultural-specific Concepts' with Jeongjagwan in terms of wearer, attendance occasion and symbolic meaning.
*Cultural-specific Concepts:* Calceus, Pileus (hat)
*Features of Jeongjagwan:* 1. Wearer: female, bride; 2. Attendance occasion: wedding; 3. Symbolic Meaning: good luck, happiness
*Features of Calceus:* 1. Wearer: female; 2. Attendance occasion: wedding, festival; 3. Symbolic Meaning: China nationalism
*Features of Pileus (hat):* 1. Wearer: female, bride; 2. Attendance occasion: wedding; 3. Symbolic Meaning: good luck, modesty
*Answer Format:* If Jeongjagwan and Calceus are more similar than Jeongjagwan and Pileus (hat) in terms of wearer, attendance occasion and symbolic meaning, please answer Calceus >Pileus (hat), otherwise answer Calceus <Pileus (hat).
*Answer:* Calceus >Pileus (hat)

*Question:* Please sort the following 'Cultural-specific Concepts' in descending order of similarity feature overlap between 'Cultural-specific Concepts' with **Honggaitou** in terms of wearer, attendance occasion and symbolic meaning.
*Cultural-specific Concepts:* **Cheongsam**, **Bridal veils**
*Features of **Honggaitou**:* 1. Wearer: female, bride; 2. Attendance occasion: wedding; 3. Symbolic Meaning: good luck, happiness
*Features of **Cheongsam**:* 1. Wearer: female; 2. Attendance occasion: wedding, festival; 3. Symbolic Meaning: China nationalism
*Features of **Bridal veils**:* 1. Wearer: female, bride; 2. Attendance occasion: wedding; 3. Symbolic Meaning: good luck, modesty
*Answer Format:* If **Honggaitou** and **Cheongsam** are more similar than **Honggaitou** and **Bridal veils** in terms of wearer, attendance occasion and symbolic meaning, please answer **Cheongsam** >**Bridal veils**, otherwise answer **Cheongsam** <**Bridal veils**.
*Answer:* |
| One-shot Prompting (w/o names) | *Question:* Please sort the following 'Cultural-specific Concepts' in descending order of similarity feature overlap between 'Cultural-specific Concepts' with Jeongjagwan in terms of wearer, attendance occasion and symbolic meaning.
*Cultural-specific Concepts:* Calceus, Pileus (hat)
*Features of Jeongjagwan:* 1. Wearer: female, bride; 2. Attendance occasion: wedding; 3. Symbolic Meaning: good luck, happiness
*Features of Calceus:* 1. Wearer: female; 2. Attendance occasion: wedding, festival; 3. Symbolic Meaning: China nationalism
*Features of Pileus (hat):* 1. Wearer: female, bride; 2. Attendance occasion: wedding; 3. Symbolic Meaning: good luck, modesty
*Answer Format:* If Jeongjagwan and Calceus are more similar than Jeongjagwan and Pileus (hat) in terms of wearer, attendance occasion and symbolic meaning, please answer Calceus >Pileus (hat), otherwise answer Calceus <Pileus (hat).
*Answer:* Calceus >Pileus (hat)

*Question:* Please sort the following 'Cultural-specific Concepts' in descending order of similarity feature overlap between 'Cultural-specific Concepts' with **concept A** in terms of wearer, attendance occasion and symbolic meaning.
*Cultural-specific Concepts:* **concept B**, **concept C**
*Features of **concept A**:* 1. Wearer: female, bride; 2. Attendance occasion: wedding; 3. Symbolic Meaning: good luck, happiness
*Features of **concept B**:* 1. Wearer: female; 2. Attendance occasion: wedding, festival; 3. Symbolic Meaning: China nationalism
*Features of **concept C**:* 1. Wearer: female, bride; 2. Attendance occasion: wedding; 3. Symbolic Meaning: good luck, modesty
*Answer Format:* If **concept A** and **concept B** are more similar than **concept A** and **concept C** in terms of wearer, attendance occasion and symbolic meaning, please answer **concept B** >**concept C**, otherwise answer **concept B** <**concept C**.
*Answer:* |

Table 7: **One-shot prompt strategies' examples.** Each instance includes a prompt template for evaluating language models, the cultural-specific concepts and the required format for responses.

| Strategy | Examples |
|---|---|
| CoT Prompting w/o features | *Question:* Please sort the following 'Cultural-specific Concepts' in descending order of similarity feature overlap between 'Cultural-specific Concepts' with Jeongjagwan in terms of wearer, attendance occasion and symbolic meaning.
*Cultural-specific Concepts:* Calceus, Pileus (hat)
*Answer Format:* If Jeongjagwan and Calceus are more similar than Jeongjagwan and Pileus (hat) in terms of wearer, attendance occasion and symbolic meaning, please answer Calceus >Pileus (hat), otherwise answer Calceus <Pileus (hat).
*Answer:* Calceus >Pileus (hat)
*Reasons:* 1. Features of Jeongjagwan: weared by man, the upper class of the Joseon period; weared in daily headgear; do not have any symbolic meaning.
Features of Calceus: weared by males in the upper-class of the Roman Republic and Empire; weared in everyday life; is the symbol of rank or social status of the wearer.
Features of Pileus (hat): weared by infantry; weared during the Saturnalia festival; is the symbols of Libertas and the goddess representing liberty.
2. Wearer: The Jeongjagwan and Calceus are both weared by the males in the upper class, the Jeongjagwan and Pileus (hat) have no common wearers' description.
3. Attendance Occasions: The Jeongjagwan and Calceus are both weared in everyday life, the Jeongjagwan and Pileus (hat) have no common occasions' description.
4. Symbolic Meaning: The Jeongjagwan do not have any common symbolic meaning with Calceus or Pileus (hat).
5. So 'Jeongjagwan' and 'Calceus' are more similar than 'Jeongjagwan' and 'Pileus (hat)', the answer is Calceus >Pileus (hat).

*Question:* Please sort the following 'Cultural-specific Concepts' in descending order of similarity feature overlap between 'Cultural-specific Concepts' with **Honggaitou** in terms of wearer, attendance occasion and symbolic meaning.
*Cultural-specific Concepts:* **Cheongsam**, **Bridal veils**
*Answer Format:* If **Honggaitou** and **Cheongsam** are more similar than **Honggaitou** and **Bridal veils** in terms of wearer, attendance occasion and symbolic meaning, please answer**Cheongsam** >**Bridal veils**, otherwise answer **Cheongsam** <**Bridal veils**.
*Answer:* |
| CoT Prompting with features | *Question:* Please sort the following 'Cultural-specific Concepts' in descending order of similarity feature overlap between 'Cultural-specific Concepts' with Jeongjagwan in terms of wearer, attendance occasion and symbolic meaning.
*Cultural-specific Concepts:* Calceus, Pileus (hat)
*Features of Jeongjagwan:* 1. Wearer: female, bride; 2. Attendance occasion: wedding; 3. Symbolic Meaning: good luck, happiness
*Features of Calceus:* 1. Wearer: female; 2. Attendance occasion: wedding, festival; 3. Symbolic Meaning: China nationalism
*Features of Pileus (hat):* 1. Wearer: female, bride; 2. Attendance occasion: wedding; 3. Symbolic Meaning: good luck, modesty
*Answer Format:* If Jeongjagwan and Calceus are more similar than Jeongjagwan and Pileus (hat) in terms of wearer, attendance occasion and symbolic meaning, please answer Calceus >Pileus (hat), otherwise answer Calceus <Pileus (hat).
*Answer:* Calceus >Pileus (hat)
*Reasons:* 1. Wearer: The Jeongjagwan and Calceus are both weared by the males in the upper class, the Jeongjagwan and Pileus(hat) have no common wearers' description.
2. Attendance Occasions: The Jeongjagwan and Calceus are both weared in everyday life, the Jeongjagwan and Pileus (hat) have no common occasions' description.
3. Symbolic Meaning: The Jeongjagwan do not have any common symbolic meaning with Calceus or Pileus (hat).
4. So 'Jeongjagwan' and 'Calceus' are more similar than 'Jeongjagwan' and 'Pileus (hat)', the answer is Calceus >Pileus (hat).

*Question:* Please sort the following 'Cultural-specific Concepts' in descending order of similarity feature overlap between 'Cultural-specific Concepts' with **Honggaitou** in terms of wearer, attendance occasion and symbolic meaning.
*Cultural-specific Concepts:* **Cheongsam**, **Bridal veils**
*Features of Honggaitou:* 1. Wearer: female, bride; 2. Attendance occasion: wedding; 3. Symbolic Meaning: good luck, happiness
*Features of Cheongsam:* 1. Wearer: female; 2. Attendance occasion: wedding, festival; 3. Symbolic Meaning: China nationalism
*Features of Bridal veils:* 1. Wearer: female, bride; 2. Attendance occasion: wedding; 3. Symbolic Meaning: good luck, modesty
*Answer Format:* If **Honggaitou** and **Cheongsam** are more similar than **Honggaitou** and **Bridal veils** in terms of wearer, attendance occasion and symbolic meaning, please answer **Cheongsam** >**Bridal veils**, otherwise answer **Cheongsam** <**Bridal veils**.
*Answer:* |
| CoT Prompting (w/o names) | *Question:* Please sort the following 'Cultural-specific Concepts' in descending order of similarity feature overlap between 'Cultural-specific Concepts' with Jeongjagwan in terms of wearer, attendance occasion and symbolic meaning.
*Cultural-specific Concepts:* Calceus, Pileus (hat)
*Features of Jeongjagwan:* 1. Wearer: female, bride; 2. Attendance occasion: wedding; 3. Symbolic Meaning: good luck, happiness
*Features of Calceus:* 1. Wearer: female; 2. Attendance occasion: wedding, festival; 3. Symbolic Meaning: China nationalism
*Features of Pileus (hat):* 1. Wearer: female, bride; 2. Attendance occasion: wedding; 3. Symbolic Meaning: good luck, modesty
*Answer Format:* If Jeongjagwan and Calceus are more similar than Jeongjagwan and Pileus (hat) in terms of wearer, attendance occasion and symbolic meaning, please answer Calceus >Pileus (hat), otherwise answer Calceus <Pileus (hat).
*Answer:* Calceus >Pileus (hat)
*Reasons:* 1. Wearer: The Jeongjagwan and Calceus are both weared by the males in the upper class, the Jeongjagwan and Pileus(hat) have no common wearers' description.
2. Attendance Occasions: The Jeongjagwan and Calceus are both weared in everyday life, the Jeongjagwan and Pileus (hat) have no common occasions' description.
3. Symbolic Meaning: The Jeongjagwan do not have any common symbolic meaning with Calceus or Pileus (hat).
4. So 'Jeongjagwan' and 'Calceus' are more similar than 'Jeongjagwan' and 'Pileus (hat)', the answer is Calceus >Pileus (hat).

*Question:* Please sort the following 'Cultural-specific Concepts' in descending order of similarity feature overlap between 'Cultural-specific Concepts' with **concept A** in terms of wearer, attendance occasion and symbolic meaning.
*Cultural-specific Concepts:* **concept B**, **concept C**
*Features of concept A:* 1. Wearer: female, bride; 2. Attendance occasion: wedding; 3. Symbolic Meaning: good luck, happiness
*Features of concept B:* 1. Wearer: female; 2. Attendance occasion: wedding, festival; 3. Symbolic Meaning: China nationalism
*Features of concept C:* 1. Wearer: female, bride; 2. Attendance occasion: wedding; 3. Symbolic Meaning: good luck, modesty
*Answer Format:* If **concept A** and **concept B** are more similar than **concept A** and **concept C** in terms of wearer, attendance occasion and symbolic meaning, please answer **concept B** >**concept C**, otherwise answer **concept B** <**concept C**.
*Answer:* |

Table 8: **Chain-of-Thought prompt strategies' examples.** Each instance includes a prompt template for evaluating large language models, the cultural-specific concepts and the required format for responses.

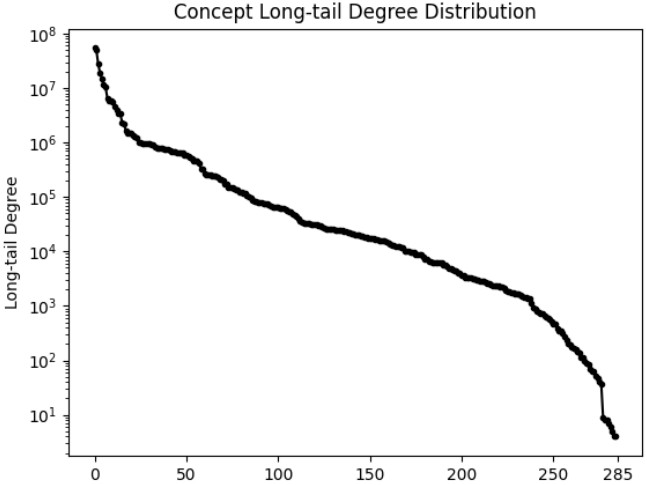

Figure 7: **The longtail degree of all cultural-specific concepts.** We calculate the long-tail degree of all cultural concepts through Google search engine.

| Category | Model | Large | | | Middle | | | Small | | |
|---|---|---|---|---|---|---|---|---|---|---|
| | | IO | One-shot | CoT | IO | One-shot | CoT | IO | One-shot | CoT |
| *Clothing* | gpt-3.5 | $50.14_{\pm 0.10}$ | $59.74_{\pm 0.64}$ | $57.23_{\pm 0.27}$ | $51.81_{\pm 0.00}$ | $55.51_{\pm 0.43}$ | $55.13_{\pm 0.11}$ | $48.66_{\pm 0.41}$ | $47.92_{\pm 0.19}$ | $51.55_{\pm 0.25}$ |
| | llama-13b | $49.13_{\pm 0.00}$ | $50.00_{\pm 0.00}$ | $50.00_{\pm 0.00}$ | $49.32_{\pm 0.00}$ | $49.77_{\pm 0.00}$ | $50.00_{\pm 0.00}$ | $49.19_{\pm 0.00}$ | $50.00_{\pm 0.00}$ | $50.00_{\pm 0.00}$ |
| | llama-7b | $36.15_{\pm 0.00}$ | $34.20_{\pm 0.00}$ | $33.19_{\pm 0.00}$ | $35.52_{\pm 0.00}$ | $36.43_{\pm 0.00}$ | $31.83_{\pm 0.00}$ | $33.60_{\pm 0.00}$ | $39.45_{\pm 0.00}$ | $33.67_{\pm 0.00}$ |
| *Food* | gpt-3.5 | $79.27_{\pm 0.15}$ | $65.49_{\pm 0.92}$ | $66.99_{\pm 0.26}$ | $54.35_{\pm 0.00}$ | $42.17_{\pm 0.31}$ | $43.11_{\pm 0.27}$ | $45.82_{\pm 0.07}$ | $31.07_{\pm 0.18}$ | $24.63_{\pm 0.21}$ |
| | llama-13b | $50.00_{\pm 0.00}$ | $50.00_{\pm 0.00}$ | $50.00_{\pm 0.00}$ | $50.00_{\pm 0.00}$ | $50.00_{\pm 0.00}$ | $50.00_{\pm 0.00}$ | $50.00_{\pm 0.00}$ | $50.00_{\pm 0.00}$ | $50.00_{\pm 0.00}$ |
| | llama-7b | $49.36_{\pm 0.00}$ | $44.55_{\pm 0.00}$ | $27.24_{\pm 0.00}$ | $48.70_{\pm 0.00}$ | $39.13_{\pm 0.00}$ | $37.61_{\pm 0.00}$ | $54.13_{\pm 0.00}$ | $46.46_{\pm 0.00}$ | $39.23_{\pm 0.00}$ |

Table 9: **The results of all experiments without concepts' name.** Based on the experiment in Table 3, concept names are removed from the prompt words.

# E   Experiments without concept names

We also conducted an experiment to remove cultural concept names to determine whether the model can make comparisons based solely on the features of the concepts, thus eliminating the influence of the concept names. Through this exploration, we hope to mitigate the long-tail effects caused by concept names. Table 9 shows the results.

# F   Error analysis

Based on our error analysis of model predictions, we identify three major types of model errors: (1) cultural knowledge errors, (2) answer format errors, and (3) concept candidate order bias. Table 10 provides an illustrative example of each error type.

Figure 8 shows the results of our error analysis. We observed that GPT-3.5 performs best in terms of answer formatting, while LLaMA-7b shows poor performance. By removing concept names in questions (see details in Appendix E), we found that both LLaMA variants (7b and 13b) are sensitive to the order of concept candidates.

| Error Type | Examples |
|---|---|
| Knowledge | **Forward prompting**
*Question:* Please sort the following 'Cultural-specific Concepts' in descending order of similarity feature overlap between 'Cultural-specific Concepts' with **Sapanpe** in terms of wearer, attendance occasion and symbolic meaning.
*Cultural-specific Concepts:* **Burzinqa, Alasho**
*Label:* **Burzinqa** < **Alasho**
*Answer:* **Burzinqa** > **Alasho**
**Reverse prompting**
*Question:* Please sort the following 'Cultural-specific Concepts' in descending order of similarity feature overlap between 'Cultural-specific Concepts' with **Sapanpe** in terms of wearer, attendance occasion and symbolic meaning.
*Cultural-specific Concepts:* **Alasho, Burzinqa**
*Label:* **Alasho** < **Burzinqa**
*Answer:* **Alasho** > **Burzinqa**
**Reasons:** The large language model maintains consistency in two different candidate concept orderings. |
| Format | *Question:* Please sort the following 'Cultural-specific Concepts' in descending order of similarity feature overlap between 'Cultural-specific Concepts' with **Honggaitou** in terms of wearer, attendance occasion and symbolic meaning.
*Cultural-specific Concepts:* **Soubane cloth, Isiagu**
*Label:* **Soubane cloth** > **Isiagu**
*Answer:* Based on the information provided, here is the comparison of the cultural-specific concepts in descending order of similarity to '**Honggaitou**' in terms of wearer, attendance occasion, and symbolic meaning: **Soubane cloth** >**Honggaitou** >**Isiagu**
**Reasons:** The large language model did not answer the question in the given format. |
| Order | **Forward prompting**
*Question:* Please sort the following 'Cultural-specific Concepts' in descending order of similarity feature overlap between 'Cultural-specific Concepts' with **Honggaitou** in terms of wearer, attendance occasion and symbolic meaning.
*Cultural-specific Concepts:* **Soubane cloth, Aso oke**
*Label:* **Soubane cloth** > **Aso oke**
*Answer:* **Soubane cloth** > **Aso oke**
**Reverse prompting**
*Question:* Please sort the following 'Cultural-specific Concepts' in descending order of similarity feature overlap between 'Cultural-specific Concepts' with **Sapanpe** in terms of wearer, attendance occasion and symbolic meaning.
*Cultural-specific Concepts:* **Aso oke, Soubane cloth**
*Label:* **Aso oke** < **Soubane cloth**
*Answer:* **Aso oke** > **Soubane cloth**
**Reasons:** Large language models answer questions in the given format, but the answer changes after swapping candidates. |

Table 10: **Examples of different error types.** Each instance is derived from real experimental data, showing different error types.

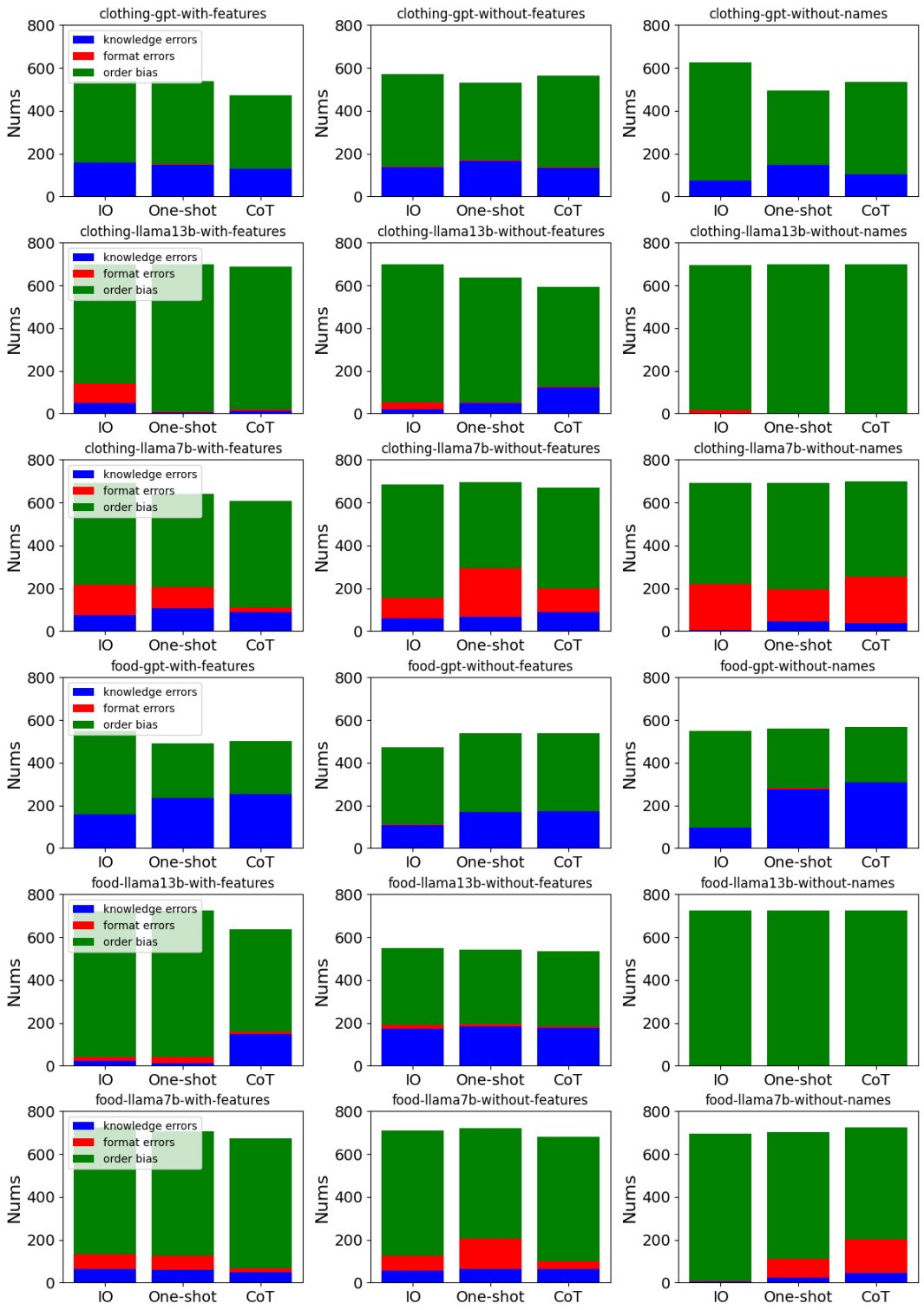

Figure 8: **Analysis of error samples of different models.** Our work conducts an error analysis of different models under different experimental groups(with features, w/o features, w/o names).

