# OpenReview forum: "How Well Do LLMs Identify Cultural Unity in Diversity?"
_colmweb.org/COLM/2024/Conference — COLM_

### Official Review · Reviewer_zPi4 · 2024-04-19

**Rating:** 6
**Confidence:** 4
**Ethics Flag:** 1

**Summary:**

There has been much research regarding whether LLM can detect or even understand cultural diversity. This paper, instead, tries to understand whether LLM is sensitive to cultural unity, i.e., the common ground across cultures. To do this, the authors introduce a new benchmark dataset CUNIT that contains evaluation examples of a variety of concepts across 10 countries.

**Questions To Authors:**

Figure 1 looks nice. However, the caption does not introduce the relations between each concept of interest itself and the features below it (And the features themselves). Therefore readers can get confused (I know they are described in later sections, but since the authors choose to place one figure on the first page to sell the paper, the authors should reduce possible confusion as possible as they can).

Table 2: unfortunately, the authors seem not to mention \mu and \delta of the difference of similarity of the final triplet samples anywhere in the paper. I believe these are very important indicators of the difficulty in different granularities.

The author mentioned that "A small level of similarity difference indicates a harder contrastive matching task". If I understand correctly from Table 2, "Small" indicates that the Sim. Difference is actually **large**, i.e., a value within ($\mu$+0.5$\delta$, 1], which should actually make the contrast task easier. It is quite confusing here. Are the authors mistaken the two granularity, i.e., "Large" and "Small"?

Figure 3: the information is redundant since the similarity of A and B is the same as the similarity of B and A. The authors can remove the upper triangular part.

**Reasons To Accept:**

- This paper is generally well-written. I appreciate the authors include many examples in the main text as well as figures, which can help the reader to better grasp the idea.

- The contribution is impactful. By benchmarking the LLMs' sensitivity to cultural unity, we can better understand the cross-lingual capability of these models.

- The description of CUNIT is quite clear in Section 3.

- The authors evaluate both LLM's results as well as human performance.

**Reasons To Reject:**

There are two limitations:

- Before evaluating LLM's sensitivity to cultural unity, I think it is necessary to first qualitatively or quantitively evaluate whether an LLM knows or understands these concepts.

- The authors did not mention the background of the two human annotators. I assume the authors want to set human performance as the upper-bound performance. However, since the concepts are across 10 countries, I am sure how well these human annotators are familiar with these concepts. So maybe a better human upper-bound performance would be to ask different people who are familiar with the concepts involved in each triplet.

Some minor suggestions are listed in the section below.

---

> ### Author Rebuttal · Authors · 2024-05-31
>
> Q1: Assessment of LLMs' perceptions on each isolated cultural-specific concept.
>
> We share your concern with the potential shortcuts learned by LLMs for prediction cross-cultural unity. Note that we have the chain-of-thought (CoT) baseline where we ask the LLM to predict a concept's cultural features and then perform our task. One easy way to assess LLMs' perceptions on each concept is to measure the accuracy of LLMs' predicted features compared to our extracted features. We show a better performance of CoT than vanilla prompting, indicating that LLMs have certain levels of perceptions on the concepts.
>
> Q2: Annotator background.
>
> Yes, the purpose of collecting human judgements is to provide the upper bounds. We agree with your thought that annotators' performance is associated with their familarity with the testing concepts. We provide the details of annotator background in the reply to Reviewer RbmH Q3.
>
> Regarding the annotation quality, we acknowledge that there may inevitabley exist potential annotation biases given the annotator's cultural background. Follow your helpful sugestions, we will broaden the annotator participants by lauching an open flatform to grow the resources of our CUNIT Benchmark.
>
> Q3: Mistaken the granularity
>
> Thanks for pointing this out. Indeed, the "Large" and "Small" in Table 2 should be flipped. We double-check our results and make sure this label mistake only happens in Table 2, and do not occurs in the other tables. We will correct this label mistake in the next version.
>
> Q4: Figure improvement.
>
> We appreciate your thoughtful feedback and will improve the figures accordingly.

---

> > ### Comment · Reviewer_zPi4 · 2024-06-05
> >
> > Thank you for your reply. I confirm that my questions have been answered. Good luck!

---

> > > ### Author Response · Authors · 2024-06-06
> > >
> > > Thanks so much for your positive feedback. We are glad that our response has addressed your questions.

---

### Official Review · Reviewer_K2wz · 2024-04-30

**Rating:** 7
**Confidence:** 4
**Ethics Flag:** 2

**Summary:**

New benchmark for identifying cross cultural similarity of traditional concepts. Through semi automatic annotation of fine grained features for food and clothing concepts, concepts similarity is defined by shared features. LLMs are evaluated on distinguishing cross cultural concepts with high similarity versus low, a novel task as a step towards cross cultural communication facilitated by LLMs.

**Ethics Concerns Details:**

No information on annotators or compensation

**Questions To Authors:**

"LLMs are still limited to capturing" should be "LLMs are still limited in their ability to capture"?

"construct this feature list at the same while annotating" - redundant "at the same time"?

In the framework of Hershcovich et al. [1], the contribution here is limited to the Common Ground dimension. What about the other dimensions? Would facilitating cross cultural communication require addressing Aboutness and Values too?

Is each matrix in figure 3 symmetric? It may be easier to read if you drop everything under the diagonal to avoid repetition.

Did you try Llama 3? It may be worth adding it to the next version. Also GPT-4.

In figure 4, "significantly" refers to what p value and what statistical test?

[1] Daniel Hershcovich, Stella Frank, Heather Lent, Miryam de Lhoneux, Mostafa Abdou, Stephanie Brandl, Emanuele Bugliarello, Laura Cabello Piqueras, Ilias Chalkidis, Ruixiang Cui, Constanza Fierro, Katerina Margatina, Phillip Rust, and Anders Søgaard. 2022. Challenges and Strategies in Cross-Cultural NLP. In Proceedings of the 60th Annual Meeting of the Association for Computational Linguistics (Volume 1: Long Papers), pages 6997–7013, Dublin, Ireland. Association for Computational Linguistics.

**Reasons To Accept:**

Well motivated, interesting and even groundbreaking goal statement, to enable cross cultural understanding and common ground facilitated by LLMs. The proposed task effectively operationalizes one aspect of this seemingly intractable goal.

Expansion of traditional cross lingual semantic matching (lexicon induction etc) to the pragmatic level by focusing on concepts used in similar contexts.

Creative and effective annotation process, where fine grained features are used to quantify concepts similarity for relatively objective quantification. There is very high agreement for the features annotation.

Interesting results showing that LLMs lag behind humans in the purposes task.

**Reasons To Reject:**

Part of the annotation process is automated with ChatGPT, but the version is not specified, and neither is any indication of the performance (quality) of that automated step. Namely, it is used to summarize Wikipedia articles to extract excerpts relevant to the defined features; but we don't know what is the recall compared to a human doing this task or if there's a performance bias towards concepts from certain countries.

While the idea of contrasive classification by number of shared features is interesting, I am not convinced it is justified to treat it as ground truth. Some concepts may be perceived by a well versed human as very similar despite having only a few shared features, maybe because some features are more important than others. And vice versa. The study here does not corroborate the calculated similarity with human judgments.

Unclear experimental setup: a setting is mentioned where concepts names are masked and only features are available, but no results for this setting are presented in the main paper. Further, it seems odd to use this setup, since it seems like it relies purely on the arithmetic ability to calculate the Jaccard index from the lists of features, and does not require any cultural awareness. This also applies to the setting where both features and concepts are provided, since high performance can simply be explained by mathematical ability.

No details about the annotators except that they are "from Eastern countries".

The feature lists seem somewhat arbitrary and are not well motivated. For example many Jewish holidays are features, though they're not necessarily very cross cultural.

---

> ### Author Rebuttal · Authors · 2024-05-31
>
> Q1: Details of ChatGPT support in feature annotation.
>
> Model: gpt-3.5-turbo-0613
>
> Quality: We conducted a preliminary performance analysis by having an annotator review the model's outputs for 50 randomly selected concepts using their full Wikipedia articles, achieving a recall of ~94.42%. To ensure high-quality feature annotation, both wiki text and model-extracted excerpts are provided.
>
> Q2: Alignment of cultural similarity measurement with human judgements.
>
> The goal of cultural similarity measure is to get the ground truth of contrastive matching on testing data. Therefore, we directly collect human answers to these cases and compare them with machine-generated ground truth. Table 3 shows a high agreement exists (~88% on average, feature given) between human judgments and ground truth.
>
> Q3: Potential arithmetic issue in feature-based LLM assessment.
>
> Thanks for pointing this out! The results of LLMs on test cases with masked concept names are in Table 9, Appendix E, and we will reference this in the main paper. We do not explicitly instruct LLMs to calculate Jaccard Index on the features for predictions, as they are still weak at calculations. The ablation study of feature-only assessment aims to compare with the concept-only one, where we find LLMs perform better with features, suggesting they may lack sufficient knowledge of a concept's cultural aspects.
>
> Q4: Annotator background.
>
> Please see our reply to Reviewer RbmH Q3.
>
> Q5: Intuition of feature extraction.
>
> We consider feature extraction mainly based on the nature of material culture[1], which is reflected in its social behavior: "the way the material is used, shared"[2]. Inspired by this, we propose three pragmatic feature categories to capture an object's cultural scenario (who, when, where) and its cultural significance (symbolic meaning). Festivals like Jewish holidays are considered as cultural scenario features.
>
> [1] Dant, T. (1999). Material culture in the social world. McGraw-Hill Education.
>
> [2] Material culture. Wikipedia.
>
> Q6: Benefits to other cultural dimensions.
>
> Our curated data, e.g., cultural significance features, can benefit the analysis of LLMs on other cultural dimensions. For example, we can explore any model biases toward certain cultural values by measuring LLM performance on concepts sharing the same cultural significance features (e.g., luck) and comparing it across different features.
>
> Q7: Citation, Writing, and Other LLMs.
>
> We will improve the manuscript accordingly.

---

### Official Review · Reviewer_Q9tF · 2024-05-11

**Rating:** 6
**Confidence:** 3
**Ethics Flag:** 1

**Summary:**

this paper proposes a framework  to analyze to what extent English-based LLMs  are able to identify cross cultural concepts,  as a way to extend the current research focusing on the ability of LLMs to be culture aware, which so far have been centered mainly in terms of linguistic variations (usage, style).
for this, the authors propose a task and a. dataset that encapsulates manually defined cross-cultural concepts and their features across ten countries, from which a probing task is designed to assess how well decoder-only LLMs detect concept with the highest cross cultural similarity. Results are somewhat expected :  i) GPT is overall winner and ii) results are highly linked to frequency across languages

**Ethics Concerns Details:**

.

**Questions To Authors:**

1.Is just focusing on English centric LLMs a bit limiting for the scope of the problem ? Moreover, it seems counterproductive as people from non-English speaking countries are not, in the first place, willing to consume or interact with information provided by an English-based LLM. In that sense, while it consider the current results relevant, are we missing a very, very big picture by not considering a multilingual setting ?

2. I think this is actually a missed opportunity to increase the contribution of the paper:
"We exclude South America due to the limited availability of its cross-cultural concepts on
Wikipedia."
As we know, most countries in South America uses Spanish as main language, but it is highly fragmented given historical and even geographical reasons. Therefore, having a single common language (Spanish), but that has clear cultural differences across more than 10 countries seems like a perfect scenario for these type of experiments (considering Spanish from Spain could even also make things more interesting as there could be an intercontinental setting)

3. i think one way of complementing the analysis and maybe to obtain more diverse results could bring known cultural structure for the analysis. For example, we know there are clusters that could be exploited , such as Korean - Japanese . Based on their historical ties and relationships (beyond their Chinese linguistic root), there are several terms in Korean that derive from Japanese . Is it possible to adapt the analysis to such setting?

**Reasons To Accept:**

the topic is really relevant in terms of the amount of personalization LLMs are expected to provide.

the dataset is represent a very valuable effort in terms of quality of annotations and expected diversity of the cross culture topics and seems to be well constructed and validated .

**Reasons To Reject:**

.

---

> ### Author Rebuttal · Authors · 2024-05-31
>
> Q1: Multilingual vs. English-centric monolingual setting.
>
> We fully agree with you about the importance of multilingual setting in cross-cultural unity. As this study is the first step initiating this research problem, we emphasize the English-centric monolingual setting, mainly because "English is the most accessible language" (as discussed in Introduction, page 2) covering diverse, high-quality cultural concepts in a wide range of geopolitical regions, especially under-represented ones (e.g., Nigeria, Thailand). We lean towards data quality compared to language coverage in our first work. A natural next step is to study a multilingual setting by translating our curated data into non-English languages or collecting more non-English data.
>
> Q2: Exclusion of cultural concepts in South America.
>
> Thanks for your insightful comment. We explored the accessibility of South American cultural concepts (i.e., clothing and food) on the Spanish Wikipedia. The number of raw clothing concepts remains limited (e.g., a maximum of 9 in Peru). After feature filtering, only 3 concepts were left. Although we could gather data from the Internet beyond Wikipedia, maintaining data quality would be challenging. To address this issue and broaden the impact of our work, we will lauch an open data curation platform that invites crowdsourcing volunteers to share their knowledge on cultural concepts and grow the valuable resources in CUNIT.
>
> Q3: Enriching cross-cultural unity analysis by known cultural structure.
>
> Our intention of exploring the influence of data-centric factors (section 5.2), especially geo-cultural proximity, on LLMs' performance, is well aligned with your consideration of the existing cultural structure in society. Particularly, we analyze GPT-3.5's performance against human annotators in the inter-/intra-group of Eastern-Western cultures. Our initial rationale for considering coarse-level cultural group proximity instead of fine-level country-based proximity is that we hope to obtain a sufficient amount of data for analysis. Despite that, we believe your suggestion of delving into some countries with strong cultural associations, particularly those with large data samples, will enrich our LLM analysis on cross-cultural unity. The idea of exploring concept-level cultural unity toward loanwords is also very inspiring. By launching the open data curation platform for collecting more data, we believe these explorations are promising to study.

---

> > ### Comment · Reviewer_Q9tF · 2024-06-05
> >
> > Thank you for the detailed answer. I think in general, the idea of having an open data curation platform as a way to continuously collect more and better data makes total sense and will definitely help in terms of discussion points 2 and 3
> >
> > I think in the case of point 1, seems a more methodological matter, I wish the authors could come up with an effective way to assess the usefulness of the proposed framework for the actual intended audience. Furthermore, I understand that for the sake of this publication, English was used as a main evaluation artifact.

---

> > ### Author Response · Authors · 2024-06-06
> > **Follow-up Response**
> >
> > Thank you so much for taking the time to read our response. We are glad that our answers have addressed some of your concerns. Regarding your point about the assessment of our framework emphasizing cultural unity, we would like to share some thoughts below.
> >
> > Given our focus on cross-cultural alignment, we can evaluate our cross-cultural similarity framework's practical benefits through downstream applications like matching translation. Following prior work on culturally-aware machine translation [1] and human translation theories [2], we learn that many culture-specific concepts in a source language lack direct equivalents in a target language. In these cases, professional human translators often use two translation strategies: (1) providing an additional explanation of the term in the target language and (2) substituting the concept with a culturally equivalent one that target language speakers can easily understand. Compared to traditional NMTS, mainly trained on parallel corpora, LLMs have a unique advantage in performing these explanation-based strategies. Therefore, our cross-cultural similarity framework could provide an initial step in assessing whether LLMs can identify culturally equivalent concepts.
> >
> > [1] Yao, B., Jiang, M., Yang, D., & Hu, J. (2023). Empowering LLM-based machine translation with cultural awareness. arXiv preprint arXiv:2305.14328.
> >
> > [2] Newmark, P. (1988). A textbook of translation (Vol. 66, pp. 1-312). New York: Prentice Hall.

---

### Official Review · Reviewer_RbmH · 2024-05-12

**Rating:** 7
**Confidence:** 4
**Ethics Flag:** 1

**Summary:**

The paper addresses the important problem of cultural awareness. It introduces a dataset CUNIT with cultural concepts and their cultural associations and similarities across 10 countries to better facilitate cultural awareness of LLMs. The authors evaluate existing LLMs on this dataset and find that they are still limited in capturing cross-cultural associations.

**Questions To Authors:**

Can you say more about the annotation process? Do these annotators come from the same cultural background of the data or they will refer to some official resources?

**Reasons To Accept:**

1. The dataset is an important contribution to the community, because it can help LLMs to create a better common ground on cross-cultural concepts to improve cultural awareness.
2.  The analysis is comprehensive and the experiment is well-designed.
3. The authors also show implication on data-centric factors and suggests the focus on long-tailed cultural knowledge.

**Reasons To Reject:**

The following are limitations general to many related problems not just to this paper in particular.
1. The evaluation method is a bit unnatural, it's a relative ranking task "sort the following cultural-specific
concepts in descending order", but this is not how these LLMs will be used in practice: they will be more used in contextualized settings like a conversation. I wonder if the paper could discuss more about how to translate this task in a more grounded setting.
2. One thing I am not sure is whether it's the best practice to quantify the similarity with an actual number and how to use this number. LLMs are not super sensitive to numeric values, and it's also hard for humans to annotate if "bridal veil" is 30% similar to "honggaitou" or 35% similar.

---

> ### Author Rebuttal · Authors · 2024-05-31
>
> Q1: Soundness of evaluation task design.
>
> We appreciate your insightful comment on the consistency between our ranking-based testing setup for LLMs and their practical applications in contextualized settings. Aligning with your consideration of LLMs' contextualized settings in practice, our task is designed in a single-turn conversational QA format. We hope to disentangle other confounding factors (e.g., context length and style) and focus on the models' capability to capture the relative cultural unity of concepts. This task could benefit downstream practices requiring "cross-cultural alignment" of concepts. For example, our task could be used to assess the cultural-centered analogical reasoning ability of an LLM-based MT system in finding appropriate replacements.
>
> Q2: Soundness of cross-cultural similarity measurement.
>
> We understand your concern with the ability of LLMs and even humans to quantify the absolute cultural similarity. To clarify, our task requires a model or human to identify a more culturally similar concept of two candidates given the query concept. Measuring the values of cultural similarity between any two concepts is only used to obtain the ground truth for the evaluation of our task.
>
> Q3: Details of human annotation.
>
> Thanks for your helpful suggestion. We share the annotation details below and will add a new section of Ethical Considerations in the paper to provide these descriptions.
>
> We have two stages of human annotation:
>
> ●Feature extraction. We employed 12 annotators from China, India, and the US to identify culturally relevant features of each concept using Wikipedia descriptions (full text and ChatGPT-highlighted excerpts). The annotators are generally bilingual (at least in English), with some being multilingual.
>
> ●Constrastive matching on testing data. To establish upper-bound performance, we gathered judgments from two additional annotators, both from China and proficient in English. Given the cultural ties in Asia, they possess sufficient cultural knowledge across Asian countries. We also provided a tutorial on our curated cultural concepts to further enrich their understanding for annotation.

---

> > ### Comment · Reviewer_RbmH · 2024-06-04
> >
> > thanks for the response

---

> > > ### Author Response · Authors · 2024-06-06
> > >
> > > Thanks a lot for your reply.

---

### Author Response · Authors · 2024-05-31

Dear Reviewer RbmH, Q9tF, K2wz, and zPi4,

Thank you all for reviewing our paper. We truly appreciate your positive feedback and detailed questions regarding our study design and experiments. Hopefully, our following explanations could address your concerns. If you have any additional questions or need further clarification, we are more than happy to engage in additional discussions.

---

### Decision · Program_Chairs · 2024-07-10

**Decision:**

Accept

**Comment:**

This paper presents a new dataset, CUNIT and an evaluation methodology for testing cultural awareness in LLMs. The evaluation dataset is created by identifying several concepts across 10 cultures (corresponding to countries) and identifying features that correspond to these concepts. Based on these features, the method calculates the cultural association between the concepts, which is used to evaluate LLMs by asking them to identify related concepts. The results suggest that LLMs still do not perform as well as humans on this task.

Overall, this work received positive comments from reviewers, who appreciated the novel task setup and dataset. They also appreciated the clarity in writing and explanation, and the fact that the authors compared LLM evaluation to human evaluation. This dataset will be useful to the community when released. There were some concerns about the two annotators who performed the feature annotation, as they may not have been familiar with all the cultural concepts spanning 10 cultures. There was also a concern about a missed opportunity of using multilingual data and not restricting to English, and the authors mentioned that it could be addressed in future work.